# Early infant diagnosis testing for HIV in a hard-to-reach fishing community in Uganda

Remegio Ndyanabo[1,2,3], Aisha Nalugya[1,3], Tonny Ssekamatte[1,3]*, Mary Nakafeero[4], Angela Kisakye[5], Aggrey David Mukose[4]

1 Department of Disease Control and Environmental Health, Makerere University School of Public Health, Kampala, Uganda, 2 Buvuma District Local Government, Lugazi, Uganda, 3 SWEEM Health Consults Limited, Kampala, Uganda, 4 Department of Epidemiology and Biostatistics, Makerere University School of Public Health, Kampala, Uganda, 5 Department of Health Policy Planning and Management, Makerere University School of Public Health, Kampala, Uganda

* ssekamattet.toca@gmail.com, tssekamatte@musph.ac.ug

## Abstract

### Background

Infants born to HIV-infected mothers are at a high risk of acquiring the infection. The World Health Organization recommends early diagnosis of HIV-exposed infants (HEIs) through deoxyribonucleic acid polymerase chain reaction (DNA PCR) and rapid HIV testing. Early detection of paediatric HIV is critical for access to antiretroviral therapy (ART) and child survival. However, there is limited evidence of the factors associated with receiving early infant diagnosis (EID) tests of the HIV testing protocol among HEIs in fishing communities in Uganda. This study established the factors associated with receiving EID tests of the HIV testing protocol among HEIs in a hard-to-reach fishing community in Uganda.

### Methods

A cross-sectional study was conducted among HEIs in selected healthcare facilities in Buvuma islands, Buvuma district. We obtained secondary data from mother-infant pair files enrolled in the EID program using a data extraction tool. Data were analysed using STATA Version 14. A modified Poisson regression analysis was used to determine the factors associated with not receiving the 1st DNA PCR test among HEIs enrolled in care.

### Results

None of the HEIs had received all the EID tests prescribed by the HIV testing protocol within the recommended time frame for the period of January 2014-December 2016. The proportion of infants that had received the 1st and 2nd DNA PCR, and rapid HIV tests was 39.5%, 6.1%, and 81.0% respectively. Being under the care of a single mother (PR = 1.11, 95% CI: 1.01–1.23, p = 0.023) and cessation of breastfeeding (PR = 0.90, 95% CI: 0.83–0.98, p = 0.025) were significantly associated with not receiving the 1st DNA PCR.

### Conclusion

Our study revealed that none of the HEIs had received all the EID tests of the HIV diagnosis testing protocol. Receiving the 1st DNA PCR was positively associated with being an infant

**Data Availability Statement:** All relevant data are within the paper and its Supporting Information files.

**Funding:** The author(s) received no specific funding for this work.

**Competing interests:** The authors have declared that no competing interests exist.

**Abbreviations:** AIDS, Acquired Immune Deficiency Syndrome; ART, Antiretroviral Therapy; DBS, Dry Blood Spot; DNA, Deoxyribonucleic Acid; EID, Early Infant Diagnosis; MTCT, Mother to Child Transmission; HIV, Human Immunodeficiency Virus; HEI, HIV Exposed Infant; MOH, Ministry of Health; PCR, Polymerase Chain Reaction; WHO, World Health Organization; UPHIA, Uganda Population-based Impact survey.

born to a single mother, and exclusive breastfeeding. Our findings highlight the need for the creation of an enabling environment for mothers and caregivers in order to increase the uptake of early diagnosis services for HEIs. Awareness-raising on the importance of EID should be scaled up in fishing communities. Demographic characteristics such as marital and breastfeeding status should be used as an entry point to increase the proportion of HEIs who receive EID tests.

## Background

HIV/AIDS is a leading cause of infant mortality in resource-limited settings worldwide [1]. Despite significant scale-up of programs to prevent mother-to-child transmission (MTCT) of HIV, over 90% of new infections among infants and young children still occur during pregnancy, childbirth, or through breastfeeding [2]. More than 150,000 children were newly infected with HIV while 100,000 died from AIDS-related causes in 2020 [1,3]. The majority of these live-in resource-limited settings in sub-Saharan Africa, where up to 30% of untreated HIV-infected children die before their first birthday, and more than 50% die before they reach 2 years of age [4]. According to the Uganda Population-based HIV Impact Assessment (UPHIA) survey of 2017, approximately 96,000 children were living with HIV and more than half (54.3%) were not on ART [2]. Survival of HIV-positive infants depends on early diagnosis and treatment [5]. Unlike in adults, disease progression is rapid in HIV-positive infants [5,6].

Owing to the risk of mortality before the age of 2 years among HIV-infected infants, the World Health Organization (WHO) recommends that national programmes should establish the capacity to provide early virological testing of HIV-exposed infants (HEIs) for HIV at six weeks or as soon as possible thereafter to guide clinical decision-making at the earliest possible stage. All infants with unknown or uncertain HIV exposure who are brought for healthcare at or around birth or at the first postnatal visit should get a 1st polymerase chain reaction (PCR) test within 6–8 weeks or the earliest opportunity thereafter followed by a 2nd PCR 6 weeks after cessation of breastfeeding. In addition, the Ugandan Ministry of Health (MoH) guidelines recommend a Dry Blood Spot (DBS) for confirmatory DNA PCR for all infants who test positive on the day they start ART; a DNA PCR test for all HEIs who develop signs and symptoms suggestive of HIV during follow-up, irrespective of breastfeeding status and a rapid HIV test at 18–24 months for all infants who test negative at 1st or 2nd PCR [7,8].

Early infant diagnosis provides an opportunity to offer optimal and timely treatment of HIV-infected children and informs decision-making on infant feeding which improves treatment outcomes [9,10]. Whereas EID is important in mitigating MTCT, its implementation has been challenging in resource-limited settings. For instance, more than two-fifths (40%) of the infants living with HIV worldwide were left undiagnosed in 2020 [3,11]. Similarly, the HIV status was unknown for nearly two-thirds (60.6%) of the children aged 0–4 years living with HIV in Uganda in 2017 [2]. This falls way below the 95-95-95 targets of diagnosing 95% of all HIV-positive individuals, enrolment of 95% of all diagnosed HIV-positive individuals on ART, and achieving 95% viral suppression for those on ART by 2030 [12,13].

Current evidence suggests that fishing communities in Uganda have a higher HIV prevalence and HIV incidence rate compared to the general population [14]. A higher HIV prevalence and incidence rate translate into a higher proportion of HEIs, thus the need to strengthen the provision of EID services in healthcare facilities. There's, however, a dearth of evidence of the factors associated with receiving EID tests of HIV testing protocol among HEIs in fishing communities in Uganda. We established the factors associated with receiving EID

tests of HIV testing protocol among HEIs in a hard-to-reach fishing community in Buvuma district, Uganda.

## Materials and methods

### Study design and area

A cross- sectional study employing quantitative data collection methods was conducted in Buvuma islands, Buvuma district, Uganda. Buvuma district is made up of 52 scattered islands in the northern shores of Lake Victoria in the central region of Uganda (Fig 1). In 2014, Buvuma islands had a population of 89,890 people, of which 48,414 were males and 41,476 were females [16]. Fishing is the major economic activity in the area. Given that the population in Buvuma is a fishing community, it is considered to be at a high risk for transmission of HIV. Available data indicate that the HIV prevalence in Buvuma islands is as high as 11.5%, and is above the national average of 6% [17]. The high HIV prevalence in this community is due to the frequent mobility of the residents, transactional and commercial sex, engagement in multiple sexual partnerships, high consumption of psychoactive substances, poor health infrastructure, and limited access to health services. To date, there are a number of interventions aimed at understanding the HIV status of HEIs. Through efforts by the government of Uganda and implementing partners, all healthcare facilities have been recommended to offer EID

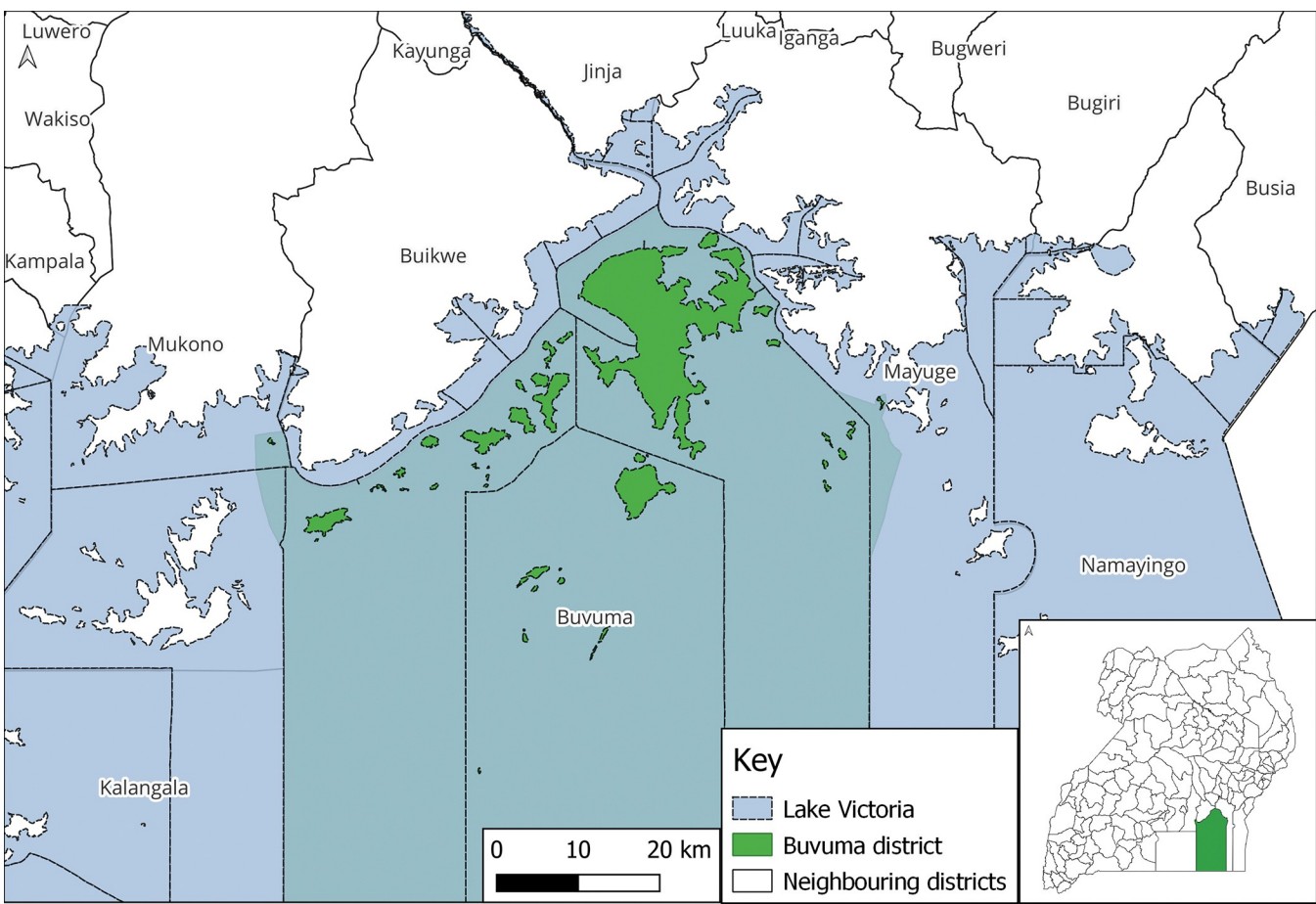

**Fig 1. Map of Buvuma district and its relative location on the map of Uganda map.**

services at both static and outreach programs such as immunization and postnatal care. The district has 12 healthcare facilities, 10 public and 2 private-not-for profit (PNFP). It has one Health centre IV, four Health centre IIIs and the rest being health centre IIs [18]. Although EID is offered in all healthcare facilities in Buvuma, mothers and care takers of HEIs find it difficult to access services due to the high transport costs, long distances to healthcare facilities and lack of partner support. Healthcare facilities also find it challenging to follow-up mothers of HEIs due to the mobile nature of the fishing population. Other challenges that limit uptake of EID testing services in this island district include lack of partner support and HIV-related stigma.

## Study population and eligibility criteria

The study population were mother-infant pairs who enrolled on the EID program between January 2014 and December 2016 in the selected healthcare facilities. This period was chosen because infants were expected to have completed the 2-year EID cascade. Mother-infant pair files with missing data on key variables (e.g. age, sex of the infant, marital status, level of education) were excluded from the sample.

## Sample size determination and sampling technique

The sample size was estimated using the Leslie Kish formula [19]. An estimated prevalence of 32% of HEIs who complete the EID testing cascade [20] was considered, the standard normal deviation at 95% confidence (1.96), and a 5% margin of error yielded a minimum sample size of 190 mother-infant pair files. Considering a missingness of files of 40% based on a study by Gloyd, Wagenaar [21]. The calculated sample size was 317. The distribution of mother-infant files per selected healthcare facility was based on sampling proportionate to size as is indicated in Table 1 below.

## Sampling procedure

Healthcare facilities were categorized based on level of service delivery. The district has one health centre IV and three health centre IIIs. We purposively selected the only health centre IV and all IIIs were selected since they are mandated to offer ART services including EID. Simple random sampling without replacement was used to select two health centre IIs out of the five for geographical representation of the island. For health centre IIs, their names were written on pieces of paper, folded and put in separate boxes depending on the level of care. The box was shaken such that they are mixed. A piece of paper was then picked at a time without replacement. At each level, two healthcare facilities were selected and the selected healthcare facility name written down.

Table 1. Distribution of mother-infant files per selected healthcare facility.

| Name of Health facility | Total number of HEIs at the Health facility (N) (2014–2016) | Sample size (n) |
|---|---|---|
| Buvuma Health Centre IV | 160 | 116 |
| Bugaya Health Centre III | 62 | 45 |
| Busamuzi Health Centre III | 60 | 44 |
| Namatale Health Centre III | 55 | 40 |
| Lwajje Health Centre II | 49 | 36 |
| Lubya Health centre II | 49 | 36 |
| **Total** | **435** | **317** |

## Theoretical framework

We used the Andersen and Newman's framework of health services utilisation to examine the factors that either facilitated or impeded utilization of EID services. According to Andersen and Newman [15], there are three key elements in the model: predisposing factors (which include, demographic characteristics, social structural variables, and an individual's basic beliefs, attitudes, and knowledge pertaining to health services), enabling (resources available, whether individually or in a community), and need-for-care factors (illnesses, conditions, and health statuses requiring health services), which either facilitate or hinder the utilization of services by individuals. The independent variables in this study included healthcare facility factors such as availability of supplies for EID, sample transportation system, turn-around time, linkage and follow up system, and availability of tracking tools; and Maternal characteristics such as socio-demographics (age, occupation, education, wealth status, marital status, location of residence (rural vs urban), ART status, mobility, stigma, HIV disclosure status to partner, place of delivery and distance from health facility.

## Study variables and measurement

The primary dependent variable was receiving all the EID tests of HIV testing protocol. This was a binary outcome which was defined as having received (yes) or not having received (No) all the three EID tests within the prescribed time frame i.e. receiving the 1st DNA PCR at 6–8 weeks, 2nd DNA PCR at 6 weeks after cessation of breast feeding and an HIV rapid test at 18–24 months. The recommended time for cessation of breastfeeding is 1 year for infants who turn HIV negative at the second PCR and up to 2 years for those who turn HIV positive at any level of the cascade. HEIs who tested HIV positive after taking any of the tests were considered to have received all the EID tests at that stage. An infant was considered to not have received all the EID tests if they did not meet the above criteria. Percentage of HEIs enrolled into care that received all the EID tests of the HIV testing protocol was calculated by dividing the total number of HEIs who had actually received the 1st and 2nd DNA PCR tests, and rapid diagnostic test within the recommended timeframe by the total number of HEIs in care and multiplying by 100. The secondary dependent variable of interest in this study was receiving the 1st DNA PCR. This was a binary outcome which was defined as having received (yes) or not having received (No) the 1st DNA PCR 6–8 weeks. An infant was considered not to have received the 1st DNA PCR if it was not done within 6–8 weeks.

## Data collection, management and analysis

A review of EID registers, mother-infant pair files, ART registers and DBS dispatch forms was done to determine the proportion of HEIs who had received EID tests according to the Uganda MoH guidelines. An electronic data extraction tool designed using the KoboCollect mobile application was used to collect information on the health facility variables as well as infant and mother characteristics. Data were field edited for consistence and omissions. Electronic data were transferred from Microsoft Excel to Stata® version 14 (Statacorp, College Station, TX) software for statistical analyses. Exploratory data analyses were conducted to check the consistency and cleanliness of data. Data were cleaned and assembled into analytic dataset. Descriptive statistics were obtained for the categorical variables (e.g., occupation, education level, marital status, location of residence, ART status, mobility, stigma, place of delivery etc) and presented as frequencies and percentages. In addition, continuous variables such as age and time to undergo EID diagnosis, were summarized using measures of central tendency and dispersion.

To establish the factors associated with not receiving the 1st DNA PCR among HEIs enrolled into care in Buvuma islands, Buvuma district, categorical variables were cross tabulated to identify the proportion of cases within a subgroup. Thereafter, bivariate analysis was conducted to determine the relationship between the independent and outcome variable. Unadjusted prevalence ratios, corresponding 95% confidence intervals and p-values were obtained using a "modified" Poisson regression. All independent variables associated with not receiving the 1st DNA PCR test at bivariate analysis with a p-value of less than 0.20 were considered for multivariable analysis. A "modified" Poisson regression analysis was used to obtain adjusted prevalence ratios given that the prevalence of not receiving the 1st DNA PCR was common (greater than 10%) [22]. During model building, a forward stepwise strategy was used [23,24]. This involved a stepwise addition of independent variables into multivariable model. After adjusting for the individual independent variables, a p value of less than 0.05 was considered statistically significant.

## Quality control and assurance

We recruited a total of 5 research assistants. In order to ensure quality control, all the research assistants received a two-day training on the study, research ethics, use of the data extraction tool, and were supervised during data collection and entry process. Pre-visiting the study area and pretesting of the study instruments was done to ensure the appropriateness of the questions for reliable and accurate information. The data abstraction tool was pre-tested from Bussi Health Centre IV located at Bussi Island, Wakiso district. This was deemed appropriate since it also serves as a hard to reach fishing community on Lake Victoria with characteristics similar to those of Buvuma islands. After the pre-test, appropriate adjustments on the tool were made before actual data collection.

## Ethical approval and consent to participate

This study was reviewed and approved by the Makerere University School of Public Health Higher Degrees and Research Ethics Committee (MakSPH HDREC). Permission to conduct this study was also sought from Buvuma District Local government. The study involved a review of patient records thus making it impractical to obtain informed consent. Nonetheless, a waiver of informed consent was granted by MakSPH REC. Each patient's records were anonymised with a unique identifier to uphold privacy and confidentiality of personal information.

## Results

A total of 435 files were screened, among these, 125 did not have information on the variables of interest (Fig 2).

### Socio-demographic characteristics

A total of 310 mother-infant pair files were reviewed. The average age of the mothers was 28.3 (SD±5.3). About 90% (281/310) of the mothers were married and 36.8% (114/310) had been diagnosed with HIV during pregnancy. Nearly two-thirds, 61.3% (190/310) had disclosed their HIV status to their sexual partners and 67.1% (208/310) had their last delivery in a health facility. Nearly half, 47.1% (146/310) of the infants included in the sample were females (Table 2).

### Sample collection and communication of EID results to mothers and caretakers

Almost all (98.1%, 304/310) the mother-infant pair files indicated that the 1st PCR sample was collected; 43.9% (132/298) indicated that the 2nd PCR sample was done while only 46.4% (137/295) indicated that the rapid diagnostic test for the infant had been done (Table 3).

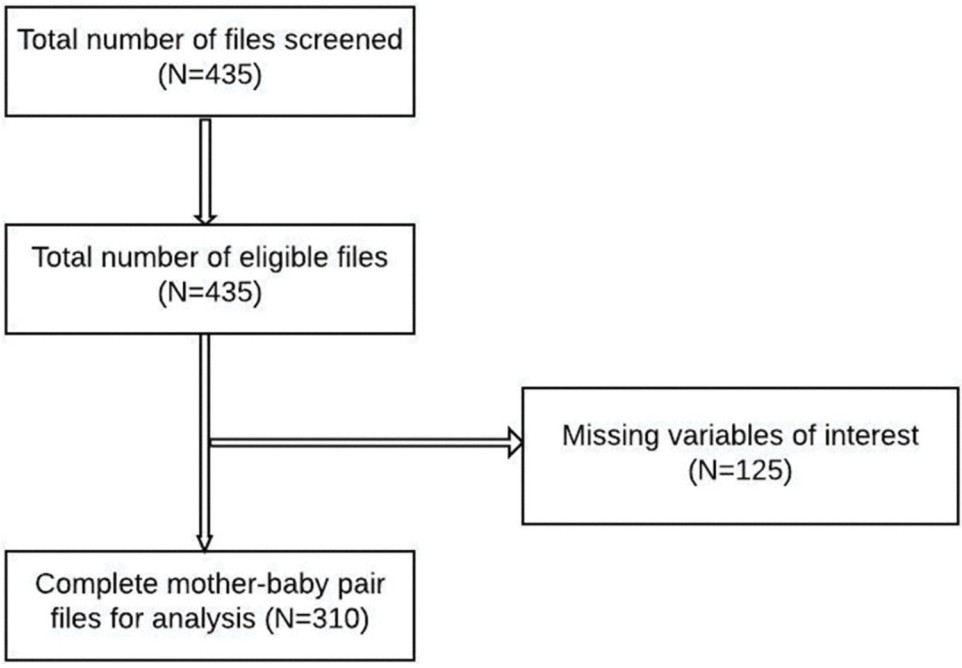

**Fig 2. Flow chart on selection of study mother-infant pair files.**

**Table 2. Socio demographic characteristics of the study participants.**

| Variable | Category | Frequency (N = 310) | Percentage (%) |
|---|---|---|---|
| **Age category of the mothers (Mean (28.3±5.3)** | Below 20 years | 16 | 5.2 |
| | 20+ years | 293 | 94.8 |
| **Marital status** | Married | 281 | 90.6 |
| | Single | 29 | 9.4 |
| **Time of HIV diagnosis for the mother** | Before pregnancy | 86 | 27.7 |
| | During antenatal care (ANC) | 114 | 36.8 |
| | During postnatal care (PNC) | 81 | 26.1 |
| | During immunisation outreaches | 29 | 9.4 |
| **Disclosed HIV status to the partner** | Yes | 190 | 61.3 |
| | No | 17 | 5.5 |
| | Missing | 103 | 33.2 |
| **Place of delivery for the infant** | Health facility | 208 | 67.1 |
| | Home | 4 | 1.3 |
| | Missing | 98 | 31.6 |
| **Distance to health facility** | ≤5kilometers | 144 | 46.5 |
| | > 5kilometers | 135 | 43.5 |
| | Missing | 31 | 10.0 |
| **Sex of infant** | Female | 146 | 47.1 |
| | Male | 158 | 51.0 |
| | Missing | 6 | 1.9 |
| **Have a treatment supporter (N = 241)** | Yes | 230 | 95.4 |
| | No | 11 | 4.6 |

**Table 3. Sample collection and communication of EID results to mothers and caretakers of during 2014–2016 in a hard-to-reach fishing community in Uganda.**

| Variable | Category | Frequency | Percentage (%) |
|---|---|---|---|
| 1st PCR sample was collected [N = 310] | Yes | 304 | 98.1 |
| | No | 6 | 1.9 |
| 2nd PCR sample was collected [N = 298] | Yes | 132 | 43.9 |
| | No | 166 | 56.1 |
| Received rapid diagnostic test [N = 295] | Yes | 137 | 46.4 |
| | Missing | 158 | 53.6 |
| Result of 1st DNA PCR | Positive | 12 | 3.9 |
| | Negative | 258 | 83.2 |
| | Missing | 40 | 12.9 |
| Result of 2nd DNA PCR | Positive | 3 | 1 |
| | Negative | 122 | 41.4 |
| | Missing | 173 | 58.6 |
| Results of 1st DNA PCR communicated | Yes | 241 | 79.3 |
| | Missing | 63 | 20.7 |
| Result of 2nd DNA PCR communicated | Yes | 120 | 40.3 |
| | Missing | 178 | 59.7 |

## Average time taken to undergo EID tests and turnaround time

The average time taken by infants to undergo the 1st PCR and 2nd PCR tests was 14.5 (SD ±16.8) and 61.0 (SD±29.1) weeks respectively, while the average time taken to undergo the rapid diagnostic test was 84.1 (SD±23.4) weeks. The average time taken from testing to receiving results (turnaround time) for the 1st and 2nd PCR was 11.4 (SD±21.1) and 11.9 (SD±35.1) weeks respectively (Table 4).

## Distribution of EID tests of the HIV testing protocol

Overall, none of the HEIs had received all the recommended tests within the recommended time frame. About 39.5% (120/304) of the HEIs had received the 1st PCR test within the recommended time of 6–8 weeks, only 6.1% (8/132) had received the 2nd PCR test at the recommended time of 58 weeks while 81.0% had received the rapid diagnostic test within the recommended time of 126–168 weeks (Table 5).

## Distribution of EID tests of HIV testing protocol stratified by demographic characteristics

A higher proportion, 41.1% (113/275) of infants of married participants received the 1st PCR test compared to 24.1% (7/29) their single counterparts. Close to half, 45.5% (65/143) of the female infants received the 1st PCR test compared to 35.5% (55/158) of the male infants. About

**Table 4. Average time taken to conduct early infant diagnosis tests and turnaround time during 2014–2016 among HIV exposed infants in a hard-to-reach fishing community in Uganda.**

| Measures of central tendency | 1st PCR testing time (weeks) | 2nd PCR testing time (weeks) | Rapid diagnostic testing time (weeks) | 1st PCR turnaround time (weeks) | 2nd PCR turnaround time (weeks) |
|---|---|---|---|---|---|
| Mean | 14.5 | 61.0 | 84.1 | 11.4 | 11.9 |
| SD | 16.8 | 29.1 | 23.4 | 21.1 | 35.1 |
| Median | 8 | 62 | 84 | 7 | 9.5 |
| N | 304 | 132 | 138 | 248 | 120 |

**Table 5. Distribution of early infant diagnosis tests of the HIV testing protocol during 2014–2016 among HIV exposed infants in a hard-to-reach fishing community in Uganda.**

| Variable | Category | Frequency | Percentage (%) |
|---|---|---|---|
| Received all the tests (N = 310) | Yes | 0 | 0.0 |
| Received 1st PCR (Between 6–8 weeks) (N = 304) | Yes | 120 | 39.5 |
| | No | 184 | 60.5 |
| Received 2nd PCR (at 58 weeks) (N = 132) | Yes | 8 | 6.1 |
| | No | 124 | 93.9 |
| Received rapid HIV test (18–24 months) (N = 140) | Yes | 111 | 81.0 |
| | No | 29 | 19.0 |

41% (40.6%, 76/187) of infants of the mothers who had disclosed their HIV status to their partners and only 31.3% (5/16) of infants of mothers who had not disclosed their HIV status to their partners had received the 1st PCR test. Nearly half, 44.7% (63/141) of the infants of clients who lived ≤5kilometers to the healthcare facility and only 36.8% (49/135) of the infants of participants who lived >5kilometers from the healthcare facility had received the 1st PCR test. In addition, only a quarter 25.0% (1/4) of the infants of the participants who delivered from home compared to 38.1% (77/202) of those who delivered from a healthcare facility had received the 1st PCR test (Table 6).

## Factors associated with not receiving the 1st DNA PCR test

The prevalence of not receiving the 1st DNA PCR test was 11% higher among infants of single mothers compared to those whose mothers were married after adjusting for time of mother

**Table 6. Distribution of 1st DNA PCR test during 2014–2016 stratified by socio-demographic characteristics of HIV exposed infants in a hard-to-reach fishing community in Uganda.**

| Variable | Category | Received N (row %) | Did not receive N (row %) | Total |
|---|---|---|---|---|
| Marital status (n = 304) | Married | 113 (41.1.) | 162 (58.9) | 275 |
| | Single | 7 (24.1) | 22 (75.9) | 29 |
| Time of HIV diagnosis for the mother (n = 304) | Before pregnancy | 33 (39.3) | 51 (60.7) | 84 |
| | During antenatal care (ANC) | 42 (37.8) | 69 (62.2) | 111 |
| | During postnatal care (PNC) | 32 (40.0) | 48 (60.0) | 80 |
| | During immunisation outreaches | 13 (44.8) | 16 (55.2) | 29 |
| Disclosed HIV status to partner (n = 203) | Yes | 76 (40.6) | 111 (59.4) | 187 |
| | No | 5 (31.3) | 11 (68.8) | 16 |
| | Total | 81 (39.1) | 126 (60.9) | 207 |
| Place of delivery (n = 206) | Health facility | 77 (38.1) | 125 (61.9) | 202 |
| | Home | 1 (25.0) | 3 (75.0) | 4 |
| | Total | 78 (36.8) | 134 (63.2) | 212 |
| Distance to the health facility (n = 274) | ≤ 5kilometers | 63 (44.7) | 78 (55.3) | 141 |
| | >5kilometers | 49 (36.8) | 84 (63.2) | 135 |
| | Total | 112 (40.1) | 167 (59.9) | 279 |
| Sex of infant (n = 304) | Female | 65 (45.5) | 78 (54.5) | 143 |
| | Male | 55 (35.5.) | 100 (64.5) | 158 |
| | Total | 120 (39.5) | 184 (60.5) | 304 |
| Type of infant feeding (N = 303) | Exclusive breast feeding | 53 (34.0) | 103 (66.6) | 156 |
| | No longer breast feeding | 56 (47.9) | 61(52.1) | 117 |
| | Complementary feeding above 6 months | 11 (36.7) | 19 (63.3) | 30 |

**Table 7. Factors associated with not receiving the 1ˢᵗ DNA PCR test during 2014–2016 among HIV exposed infants in a hard-to-reach fishing community in Uganda.**

| Variable | Number | Received 1st DNA PCR | | Unadjusted PR (95% CI) | p-value | Adjusted PR (95% CI) | p-value |
|---|---|---|---|---|---|---|---|
| | | Yes | No | | | | |
| **Marital status** | | | | | | | |
| Married | 275 | 113 (94.2) | 162 (88.0) | 1.0 | | 1.0 | |
| Single | 29 | 7 (5.8) | 22 (12.0) | 1.10 (1.00–1.21) | 0.038 | 1.11(1.01–1.23) | 0.023 |
| **Time of mother diagnosis** | | | | | | | |
| Before pregnancy | 84 | 33 (27.5) | 51 (27.7) | 1.0 | | | |
| During antenatal care (ANC) | 111 | 42 (35.0) | 69(37.5) | 1.00 (0.92–1.09) | 0.837 | | |
| During postnatal care (PNC) | 80 | 32 (26.7) | 48 (26.1) | 0.99 (0.90–1.09) | 0.926 | | |
| During immunisation outreaches | 29 | 13 (10.8) | 16 (8.7) | 0.96 (0.84–1.10) | 0.607 | | |
| **HIV disclosure status (n = 207)** | | | | | | | |
| Yes | 187 | 76 (93.8) | 111 (91.0) | 1.0 | | | |
| No | 16 | 5 (62.2) | 11 (11.0) | 1.05 (0.91–1.22) | 0.429 | | |
| **Place of infant delivery (n = 212)** | | | | | | | |
| Health facility | 202 | 77(98.7) | 125 (97.7) | 1.0 | | | |
| Home | 4 | 1 (1.3) | 3 (2.3) | 1.08 (0.844–1.38) | 0.536 | | |
| **Distance to facility (n = 279)** | | | | | | | |
| ± 5kilometers | 141 | 63 (56.3) | 78 (48.1) | 1.0 | | 1.0 | |
| > 5kilometers | 133 | 49 (43.7) | 84 (51.9) | 1.05 (0.97–1.13) | 0.186 | 1.04 (0.86–1.12) | |
| **Sex of the infant (n = 304)** | | | | | | | |
| Female | 143 | 65 (54.2) | 78 (43.8) | 1.0 | | 1.0 | |
| Male | 155 | 55 (45.8) | 100 (56.2) | 1.06 (0.99–1.14) | 0.08 | 1.06 (0.99–1.14) | 0.088 |
| **Infant feeding** | | | | | | | |
| Exclusive breast feeding | 156 | 53 (44.2) | 103 (56.3) | 1.0 | | 1.0 | |
| Cessation of breast feeding | 96 | 51 (42.5) | 45 (24.6) | 0.88 (0.81–0.95) | 0.003 | 0.90 (0.83–0.98) | 0.028 |
| Complementary feeding above 6 months | 30 | 11 (9.2) | 19 (10.4) | 0.98 (0.87–1.10) | 0.78 | 1.01 (0.90–1.15) | 0.753 |
| Never breast fed | 21 | 5 (4.2) | 16 (8.7 | 1.06 (0.94–1.18) | 0.302 | 1.09 (0.96–1.24) | 0.163 |

diagnosis for HIV, HIV disclosure status, place of delivery, distance to healthcare facility, sex of the infant and method of infant feeding (PR 1.11, 95% CI: 1.01–0.23, p = 0.023). The prevalence of not receiving the 1ˢᵗ DNA PCR test was 10% lower among infants of mothers who were no longer breast feeding compared to those of mothers practicing exclusive breast feeding after adjusting for time of mother diagnosis for HIV, HIV disclosure status, place of delivery, distance to healthcare facility, sex of the infant (PR 0.90, 95% CI: 0.83–0.98, p = 0.028). There was a borderline statistical significance between the sex of the infant and not receiving the 1ˢᵗ DNA PCR test. The prevalence of not receiving the 1ˢᵗ PCR test was 6% higher among male infants compared the females (PR 1.06, 95% CI: 0.99–1.14, p = 0.088) (Table 7).

## Discussion

This study established the factors associated with not receiving early EID tests of HIV testing protocol among HEIs in a hard-to-reach fishing community in Buvuma district, Uganda. The study revealed that none of the HEIs had received all the recommended EID tests within the prescribed timeframe. Only 39.5% of the infants had received the 1st PCR test within the recommended time of 6–8 weeks, only 6.1% had received the 2nd PCR test at the recommended

time of 58 weeks and 81.0% had received the rapid diagnostic test within the recommended time of 126–168 weeks. Being born to a single mother and not exclusively being breastfed were associated with not receiving the 1st DNA PCR.

Although none of the infants had received all the tests within the recommended timeframe, a significant proportion of mothers had completed the 1st PCR within the recommended time-frame. This may be so because the 1st PCR test is conducted at 6–8 weeks, a period when mothers are supposed to take their infants for immunisation based on the Ugandan immunisation schedule and also attend postnatal care [7,8]. However, a lower proportion of the HEIs had completed the 2nd PCR test within the recommended time frame. This could be attributed to the fact that the 2nd PCR is conducted at 58 weeks, shortly after the completion of the immunisation schedule. The fact that mothers struggle to access health care facilities for services means that bringing their infants for the 2nd PCR test 6 weeks after immunisation is burdensome. Low proportions of HEIs who have undergone the 2nd DNA PCR test have also been reported in other parts of Uganda [25,26].

Our study revealed that the prevalence of not receiving the 1st DNA PCR test was higher among infants of single mothers compared to the married. This could be to the fact that single mothers have limited support for transport needs as well as food and reminders to go to the healthcare facility. These have been shown to affect the mothers' motivation towards taking their infants for EID services. This is in line with the findings of Bwana et al., (2016) who reported that lack of partner support hindered receiving prescribed EID. Our findings indicate a need to sensitize mothers, especially those who are single on the significance of EID. In addition, the study found that the prevalence of not receiving 1st DNA PCR test was lower among infants of mothers who were no longer breast feeding compared to those of mothers practicing exclusive breast feeding. This could be attributed to the reduced bonding between the mother and the infant, and the fact that mother's assumption that their infants cannot contract the virus once not breast feeding.

Our study revealed that only 31.3% of infants of clients who had not disclosed their HIV status to their partners had received the 1st PCR test compared to 40.6% of infants of clients who had disclosed their HIV status to their partners. This could be because disclosure of HIV status to partners is associated with spousal support in the form of money to facilitate transport to the healthcare facility, and reminders and accompaniment to the healthcare facility for the services. Consequently, this could influence the distribution of EID tests among HEIs. Our findings concur with those reported in Uganda which showed that HIV status disclosure by women to partners was associated with increased spousal support and increased visits to healthcare facilities [27,28]. This indicates a need for encouraging disclosure of HIV status in order to enhance uptake of EID testing services.

Nearly half (44.7%) of the infants of clients who lived ≤5kilometers from the healthcare facility and only 36.8% of the infants of clients who lived >5kilometers from the healthcare facility had received the 1st PCR test. Longer distance from the healthcare facility implies that the mother would incur higher transportation costs, which could in turn impede uptake of EID testing services. Similarly, Samson, Mpembeni [29] in a study conducted to assess the uptake of EID at six weeks after cessation of breastfeeding among HEIs in Tanzania reported that the most common reasons for non-uptake of the test mentioned by respondents were long distance from home to the healthcare facility (22.9%, 95% CI: 20.3–25.4).

## Strengths and limitations

This study may have been the first to examine the factors associated with receiving EID tests among HEIs in a hard-to-reach fishing community. Therefore, it provides useful insights into

utilisation of EID testing services in such settings. The study utilises a relatively large sample size which makes the findings generalizable in a similar setting. This study was, however, affected by missing data in the patient files. Nonetheless, missingness of data had been catered for in the sample size calculation. Furthermore, findings from this study can only be applied to hard-to-reach fishing communities and may not be applicable to other hard-to- reach populations especially communities that are not mobile.

## Conclusions

Our study revealed that none of the HEIs had received all the EID tests of the HIV testing protocol during January 2014 to December 2016. A significant proportion of HEIs undertook timely rapid diagnostic test, while a few undertook timely 1st and 2nd DNA PCR tests. Slightly more than a third of the HEIs had undergone the 1st DNA PCR test, less than a tenth had undergone the 2nd DNA PCR test, and more than three quarters of HEIs had at least received the HIV rapid diagnostic test. Cessation of breast feeding and being under the care of non-married females were found to be associated with not receiving the 1st DNA PCR test. This study suggests the need to create awareness of the importance of adhering to the EID testing protocol among mothers, with more emphasis put on single mothers and those that cease breastfeeding.

## Supporting information

**S1 Dataset.**
(XLSX)

**S1 File.**
(PDF)

## Acknowledgments

Our sincere gratitude goes to the research assistants that included Patience Oputan, Berna Nakiyagga and Jimmy Masereka for their tireless efforts during the data collection process. We thank Dr. John Bosco Isunju (Makerere University School of Public Health) for generating the map presented in this manuscript. Special thanks also go out to the healthcare providers of Buvuma District local government for providing the records from which the data were extracted.

## Author Contributions

**Conceptualization:** Remegio Ndyanabo, Aisha Nalugya, Tonny Ssekamatte, Angela Kisakye, Aggrey David Mukose.

**Formal analysis:** Remegio Ndyanabo, Aisha Nalugya, Mary Nakafeero, Angela Kisakye, Aggrey David Mukose.

**Investigation:** Remegio Ndyanabo, Aisha Nalugya, Tonny Ssekamatte, Angela Kisakye, Aggrey David Mukose.

**Methodology:** Remegio Ndyanabo, Aisha Nalugya, Tonny Ssekamatte, Mary Nakafeero, Angela Kisakye, Aggrey David Mukose.

**Project administration:** Remegio Ndyanabo, Tonny Ssekamatte, Mary Nakafeero.

**Resources:** Remegio Ndyanabo.

**Supervision:** Remegio Ndyanabo, Angela Kisakye, Aggrey David Mukose.

**Writing – original draft:** Remegio Ndyanabo, Aisha Nalugya, Tonny Ssekamatte, Mary Nakafeero, Angela Kisakye, Aggrey David Mukose.

**Writing – review & editing:** Remegio Ndyanabo, Aisha Nalugya, Tonny Ssekamatte, Mary Nakafeero, Angela Kisakye, Aggrey David Mukose.

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
