## [Decision Letter · Decision Letter 0]

17 Nov 2022

PONE-D-22-12468Adherence to the HIV early infant diagnosis testing protocol among HIV exposed infants in a hard-to-reach fishing community in UgandaPLOS ONE

Dear Dr. Ssekamatte,

Thank you for submitting your manuscript to PLOS ONE. After careful consideration, we feel that it has merit but does not fully meet PLOS ONE’s publication criteria as it currently stands. Therefore, we invite you to submit a revised version of the manuscript that addresses the points raised during the review process.

The manuscript has been evaluated by two reviewers, and their comments are available below.

The reviewers have raised a number of major concerns, specifically that the language used may not be fully appropriate for a study including minors. Could you please carefully revise the manuscript to address all comments raised?

We look forward to receiving your revised manuscript.

Kind regards,

Alice Coles-Aldridge

Editorial Office

PLOS ONE

Journal Requirements:

2. In the ethics statement in the Methods and online submission information, please ensure that you have specified what type you obtained (for instance, written or verbal, and if verbal, how it was documented and witnessed). If your study included minors, state whether you obtained consent from parents or guardians. If the need for consent was waived by the ethics committee, please include this information.

Reviewers' comments:

Reviewer's Responses to Questions

**Comments to the Author**

1. Is the manuscript technically sound, and do the data support the conclusions?

Reviewer #1: Yes

Reviewer #2: Yes

2. Has the statistical analysis been performed appropriately and rigorously? 

Reviewer #1: Yes

Reviewer #2: Yes

3. Have the authors made all data underlying the findings in their manuscript fully available?

Reviewer #1: Yes

Reviewer #2: Yes

4. Is the manuscript presented in an intelligible fashion and written in standard English?

Reviewer #1: Yes

Reviewer #2: Yes

5. Review Comments to the Author

Reviewer #1: Title: Adherence to the HIV early infant diagnosis testing protocol among HIV exposed infants in a hard-to-reach fishing community in Uganda

Overview and general comments: A well written and important article describing adherence HIV testing protocols for early infant diagnosis in high burden fishing communities in Uganda. Authors used a cross sectional study design and secondary data to determine factors associated with non-adherence to EID testing protocols. In general, the paper is well written with some areas for clarification. The language used throughout the paper seems to assume as if children were adult participants. The word adherence to EID seems inappropriate.

Specific comments: Below are a few specific comments.

Abstract: Very well written but includes a couple of sentences that should be edited for clarity.

Line 44. None of the HIV-exposed infants had “done” all the EID tests... Should be rewritten to read as “None of the HIV-exposed infants had “received” all the EID tests…

Line 51. None of the HIV-exposed infants “adhered” to all the EID tests of HIV testing protocol. Should be rewritten to read…None of the HIV-exposed infants “received” all the EID tests of HIV testing protocol.

Live 51-53. The sentence “Adherence to the 1st DNA PCR was positively associated with being a single mother and exclusive breast” should be rewritten to read as “Receiving the 1st DNA PCR was positively associated with being an infant born to a single mother and exclusively breast fed”

Introduction: Well written and gives the context under which the study was conducted. Author should consider moving part of the last paragraph to the methods section.

Methods: Describes the setting clearly but apart from fishing, no other risk or contextual factors such as sex work, transactional work, challenges accessing health care including the limited number of facilities and expense of travelling to them. Lack of electricity. Describing these will help readers understand the results.

Sample size. Not sure of the formular is needed. The word “deviate” should be written as “deviation”.

The word adherence in methods should probably be changed especially as the paper describes infants.

Data management: under data management, some items are described as very generically. Categorical and continuous variables could be named.

Results: The data is clearly presented and linked. First title “Flow chart for mother-infant file…” is incomplete and probably not required.

Some of the description of results should be framed correctly since the study subjects were infants. For example, the line that reads, “Overall, none of the study participants had done all the recommended tests within the recommended time frame”. Could be written as none of the infants had received all recommended tests

Tables are very clear but are incompletely labeled. Each table should be labeled in such a way that it can be understood, should include the study area e.g. Buvuma and time period e.g. YYYY-YYYY

For example, table 5 is not well described. “ Table 5: Distribution of adherence to EID of HIV testing protocol at scheduled time points. Authors should rename all tables for clarity.

The subtitle after table 5, has grammatical errors and should be rewritten.

Discussion:

The last sentence of the first paragraph is not easy to understand.

Conclusion: well written.

Reviewer #2: The manuscript is written well, the methodology is clear, and the analysis was made properly and support the manuscript conclusion.

I would suggest few comments:

In section materials and methods: Providing a map illustrating the localities of the study would be more informative

In results: Numbers under subtitle "Average time taken to undergo EID tests and turnaround time" should be followed by weeks

In conclusions: Replace " marital status" by " non married females"

6. PLOS authors have the option to publish the peer review history of their article (what does this mean?). If published, this will include your full peer review and any attached files.

Reviewer #1: No

Reviewer #2: **Yes: **Mohammed A. AboElkhair

---

## [Author Response · Author response to Decision Letter 0]

20 Apr 2023

Title: Adherence to the HIV early infant diagnosis testing protocol among HIV exposed infants in a hard-to-reach fishing community in Uganda

Reviewer 1

Overview and general comments: A well-written and important article describing adherence HIV testing protocols for early infant diagnosis in high-burden fishing communities in Uganda. Authors used a cross sectional study design and secondary data to determine factors associated with non-adherence to EID testing protocols. In general, the paper is well written with some areas for clarification. The language used throughout the paper seems to assume as if children were adult participants. The word adherence to EID seems inappropriate. 

Specific comments: Below are a few specific comments.

Abstract: Very well written but includes a couple of sentences that should be edited for clarity.

Comment: Line 44. None of the HIV-exposed infants had “done” all the EID tests... Should be rewritten to read as “None of the HIV-exposed infants had “received” all the EID tests…

Response: Thank you. This has been rewritten. Page 2 Line 46

Comment: Line 51. None of the HIV-exposed infants “adhered” to all the EID tests of HIV testing protocol. Should be rewritten to read…None of the HIV-exposed infants “received” all the EID tests of HIV testing protocol.

Response: Thank you. This has been rewritten. Page 2 Line 46

Comment: Live 51-53. The sentence “Adherence to the 1st DNA PCR was positively associated with being a single mother and exclusive breast” should be rewritten to read as “Receiving the 1st DNA PCR was positively associated with being an infant born to a single mother and exclusively breast fed”

Response: Thank you. This has sentence has been rewritten. Page 2 Line 54-55

Comment: Introduction: Well written and gives the context under which the study was conducted. Author should consider moving part of the last paragraph to the methods section.

Response: Thank you. The description of the context has been moved to the methods section. Page 5 Lines 113-116

Comment: Methods: Describes the setting clearly but apart from fishing, no other risk or contextual factors such as sex work, transactional work, challenges accessing health care including the limited number of facilities and expense of travelling to them. Lack of electricity. Describing these will help readers understand the results.

Response: Contextual factors have been described. Lines 113-116. Page 5 Line 121-126

Comment: Sample size. Not sure of the formular is needed. The word “deviate” should be written as “deviation”.

Response: Sample size formular has been deleted. Page 6 Lines 137-143

Comment: The word adherence in methods should probably be changed especially as the paper describes infants.

Response: The word adherence has been dropped from the paper.

Comment: Data management: under data management, some items are described as very generically. Categorical and continuous variables could be named. 

Response: Categorical and continuous variables have been named. Page 8 Lines 193-194

Comment: Results: The data is clearly presented and linked. First title “Flow chart for mother-infant file…” is incomplete and probably not required.

Response: First title has been removed. Page 10 Lines 226-227

Comment: Some of the description of results should be framed correctly since the study subjects were infants. For example, the line that reads, “Overall, none of the study participants had done all the recommended tests within the recommended time frame”. Could be written as none of the infants had received all recommended tests

Response: This has been changed throughout the paper.

Comment: Tables are very clear but are incompletely labeled. Each table should be labeled in such a way that it can be understood, should include the study area e.g. Buvuma and time period e.g. YYYY-YYYY. For example, table 5 is not well described. “ Table 5: Distribution of adherence to EID of HIV testing protocol at scheduled time points. Authors should rename all tables for clarity. 

Response: Thank you. The captions have been updated for all tables.

Comment: The subtitle after table 5, has grammatical errors and should be rewritten.

Response: The caption has been edited. Page 13 Line 263-264

Discussion: 

Comment: The last sentence of the first paragraph is not easy to understand.

Response: Thank you. This sentence has been edited. Lines 298-299.

Comment: Conclusion: well written. 

Response: Thank you. 

 

Reviewer 2

The manuscript is written well, the methodology is clear, and the analysis was made properly and support the manuscript conclusion.

I would suggest few comments:

Comment: In section materials and methods: Providing a map illustrating the localities of the study would be more informative

Response: A map showing the localities has been provided. The map has been drawn using the shape files of administrative boundaries in Uganda. We downloaded the shape files from Uganda Bureau of Statistics.

Comment: In results: Numbers under subtitle "Average time taken to undergo EID tests and turnaround time" should be followed by weeks

Response: The word weeks has been inserted in that subsection. Lines 233-237

Comment: In conclusions: Replace " marital status" by " non married females"

Response: Thank you. The sentence has been edited. Line 341

---

## [Decision Letter · Decision Letter 1]

17 May 2023

Early infant diagnosis testing for HIV in a hard-to-reach fishing community in Uganda

PONE-D-22-12468R1

Dear Tonny Ssekamatte,

We’re pleased to inform you that your manuscript has been judged scientifically suitable for publication and will be formally accepted for publication once it meets all outstanding technical requirements.

Kind regards,

Hamufare Dumisani Dumisani Mugauri, Ph.D. Public Health

Academic Editor

PLOS ONE

Reviewer's Responses to Questions

**Comments to the Author**

1. If the authors have adequately addressed your comments raised in a previous round of review and you feel that this manuscript is now acceptable for publication, you may indicate that here to bypass the “Comments to the Author” section, enter your conflict of interest statement in the “Confidential to Editor” section, and submit your "Accept" recommendation.

Reviewer #1: All comments have been addressed

Reviewer #2: All comments have been addressed

2. Is the manuscript technically sound, and do the data support the conclusions?

Reviewer #1: Yes

Reviewer #2: Yes

3. Has the statistical analysis been performed appropriately and rigorously? 

Reviewer #1: Yes

Reviewer #2: (No Response)

4. Have the authors made all data underlying the findings in their manuscript fully available?

Reviewer #1: Yes

Reviewer #2: Yes

5. Is the manuscript presented in an intelligible fashion and written in standard English?

Reviewer #1: Yes

Reviewer #2: Yes

6. Review Comments to the Author

Reviewer #1: An excellent manuscript of great importance. The revised manuscript reads very well. All previous comments have been satisfactorily addressed

Reviewer #2: (No Response)

7. PLOS authors have the option to publish the peer review history of their article (what does this mean?). If published, this will include your full peer review and any attached files.

Reviewer #1: **Yes: **Moses Bateganya

Reviewer #2: No

---

## [Editor Report · Acceptance letter]

26 May 2023

PONE-D-22-12468R1 

Early infant diagnosis testing for HIV in a hard-to-reach fishing community in Uganda 

Dear Dr. Ssekamatte:

I'm pleased to inform you that your manuscript has been deemed suitable for publication in PLOS ONE. Congratulations! Your manuscript is now with our production department. 

Kind regards, 

on behalf of

Mr Hamufare Dumisani Dumisani Mugauri 

Academic Editor

PLOS ONE